# Unlocking the Skin-Protective Effects of *Ferulago* spp. Essential Oils: A Focus on Antidermatophytic and Wound-Healing Potential

**DOI:** 10.3390/antibiotics14040343

**Published:** 2025-03-27

**Authors:** Ceyda Sibel Kılıç, Inês Catarino, Jorge Alves-Silva, Betül Demirci, Damla Kırcı, Lígia Salgueiro, Mónica Zuzarte

**Affiliations:** 1Ankara University, Faculty of Pharmacy, Department of Pharmaceutical Botany, 06560 Ankara, Türkiye; erdurak@pharmacy.ankara.edu.tr; 2Univ Coimbra, Coimbra Institute for Clinical and Biomedical Research (iCBR), Faculty of Medicine, Azinhaga de S. Comba, 3000-548 Coimbra, Portugal; uc2024232528@student.uc.pt (I.C.); jmasilva@fmed.uc.pt (J.A.-S.); 3Univ Coimbra, Center for Innovative Biomedicine and Biotechnology (CIBB), 3000-548 Coimbra, Portugal; 4Univ Coimbra, Faculty of Pharmacy, Azinhaga de S. Comba, 3000-548 Coimbra, Portugal; ligia@ff.uc.pt; 5Clinical Academic Centre of Coimbra (CACC), 3000-548 Coimbra, Portugal; 6Anadolu University, Faculty of Pharmacy, Department of Pharmacognosy, 26470 Tepebaşı, Eskişehir, Türkiye; bdemirca@anadolu.edu.tr; 7İzmir Katip Çelebi University, Faculty of Pharmacy, Department of Pharmacognosy, 35620 Çiğli, İzmir, Türkiye; damla.kirci@ikc.edu.tr; 8Univ Coimbra, Chemical Engineering and Renewable Resources for Sustainability (CERES), Department of Chemical Engineering, 3030-790 Coimbra, Portugal

**Keywords:** dermatophytosis, biofilm, virulence, cell migration

## Abstract

Background/Objectives: Fungal infections have significant health risks due to virulence factors like biofilms that lead to chronic and persistent infections. To decrease associated rates of morbidity and mortality, it is crucial to develop effective antifungal treatments. The present study aims to evaluate the bioactive potential of *Ferulago* spp. essential oils by assessing their effect on dermatophytes and wound healing, as these fungi are often associated with wound infection. Methods: The minimal inhibitory concentration (MIC) and the minimal lethal concentration (MLC) of five essential oils were assessed on planktonic dermatophytes and the most promising were used to evaluate their effect on the formation and disruption of biofilms, by quantifying biofilm mass using crystal violet and extracellular matrix deposition using safranin staining. Alterations in fungal morphology were confirmed by optical microscopy and a cell migration assay was used to assess wound-healing capacity. Results: *Ferulago silaifolia* essential oil characterized by high amounts of α-pinene (45.4%) and *cis*-crysanthenyl acetate (39.1%) was the most active, particularly against *Microsporum canis* and *Trichophyton rubrum* (MIC = 50 µg/mL). Regarding biofilm assays, *Trichophyton rubrum* was the most susceptible strain, with both biofilm mass and extracellular matrix being highly compromised with an evident decrease in hyphal growth and mycelial density. In addition, this essential oil significantly increased fibroblast migration at 25 μg/mL, indicating a wound-healing effect that could prevent systemic infections. Conclusions: The present study provides new insights into the treatment of dermatophytosis by highlighting the antivirulent effects of *F. silaifolia* essential oil and its potential wound-healing properties.

## 1. Introduction

The Global Action Fund for Fungal Infections (GAFFI) estimates that over 6 million individuals develop life-threatening fungal infections annually, of which 3.75 million do not survive [1]. Indeed, fungal infections are becoming a worldwide public health issue, especially in immunocompromised individuals, including patients with chronic lung disease, prior tuberculosis, HIV, cancer and diabetes mellitus [2]. These infections can manifest in various forms, ranging from localized cutaneous to invasive and life-threatening fungal infections [3].

The present study focuses on dermatophytes that are known to present an infection rate of around 20–25% worldwide [4] and are divided into nine genera: *Arthroderma*, *Lopophyton*, *Nannizia*, *Ctenomyces*, *Guarromyces*, *Paraphyton*, *Microsporum*, *Epidermophyton*, and *Trichophyton*, the last three being the ones that can cause infections in humans [5,6]. These filamentous fungi affect the keratin present in the skin, hair, and nails, leading to diseases known as dermatophytosis and onychomycosis (nails), which are the the most disseminated type of mycosis. Dermatophytes represent a category of primary pathogenic fungi that rely on keratin cleavage for nutritional sustenance [3].

In the early stages of infection, dermatophytes prompt keratinocytes to generate various cytokine patterns, depending on their classification, that mediate the inflammatory response and the accumulation of leukocytes in the infected tissue [3]. Dermatophytes can be categorized as anthropophilic, geophilic or zoophilic species, depending on their normal habitat (humans, soil or animals, respectively). Clinically, dermatophytosis is classified according to the site of infection as follows: tinea capitis (affects the scalp and hair), tinea faciei (affects the face), tinea barbae (affects the beard), tinea corporis (affects the body), tinea manuum (affects the hands), tinea cruris (affects the groin), tinea pedis (affects the feet, known as athlete’s foot) and tinea unguium (affects the nails). Other clinical variants include tinea imbricata, pseudoimbricata, and Majocchi’s granuloma, the latter being mostly caused by *Trichophyton rubrum* and occurring when a long-standing superficial fungal infection causes progressive dissemination into subcutaneous tissue [7].

An important virulence factor in dermatophytes is their ability to form biofilms, a more resistant structure than planktonic forms, formed by a larger group of fungal cells and extracellular matrix. The fungal cells first adhere to a substrate, either a mucosal or abiotic surface, and proliferate until they form an extracellular matrix composed of polysaccharides, lipids, proteins, and nucleic acids. The main function of this organization is to increase fungi survival, leading to a strong resistance to conventional therapy and host defenses [8].

The most commonly used treatments for dermatophytosis include topical and oral agents. Topical antifungal agents like butenafine, clotrimazole and miconazole are recommended for dermatophytosis that presents as localized skin infections. However, oral drugs such as terbinafine, itraconazole, griseofulvin and fluconazole are recommended for more extensive infections, especially in combination therapy, to prevent the emergence of resistance and ensure wide coverage [9]. Nevertheless, despite the use of numerous antifungal drugs, addressing these infections remains a challenge, as virulence factors such as biofilms are known to confer high resistance [8]. Furthermore, it is commonplace for patients to neglect and discontinue treatment, primarily due to its long-term duration. These barriers to conventional treatments have driven the emergence of resistant strains, highlighting the need for effective alternatives such as plant metabolites like essential oils (EOs). These volatile extracts have been the target of many researchers as alternatives and/or complementary strategies in healthcare.

The genus *Ferulago* W.D.J.M. Koch (Apiaceae) is considered the eighth largest botanical family in Türkiye. Of the 48 taxa in this genus [10], 35 grow naturally in this country and 19 are endemic, thus positioning Türkiye as the gene center for *Ferulago* [11]. Several *Ferulago* species have been traditionally used for a variety of medicinal purposes, including the treatment of skin wound infections, ulcers, snake bites, headaches, and spleen problems. They are also used as sedatives, immunostimulants, carminatives, and remedies for peptic issues, intestinal worms, hemorrhoids, and even as aphrodisiacs. Moreover, some species are used as spices and to increase the shelf life of vegetable oils and dairy products [12,13]. Although the antimicrobial properties of several *Ferulago* species have been identified, regarding fungal infections, the majority of studies focus on the effect of *Ferulago* EOs on *Candida* spp. growth [14,15]. Indeed, as far as we know, only one study assessed the antifungal effect of *F. capillaris* EO against dermatophytes [16] but failed to address its impact on relevant virulence factors such as biofilms. Therefore, the present study aims to evaluate the antifungal potential of five *Ferulago* species on planktonic dermatophytes, selecting the most promising one to assess its ability to eradicate and/or prevent biofilm formation and promote wound healing.

## 2. Results

### 2.1. Chemical Composition

Table 1 summarizes the main chemical compounds found in the EOs obtained from different species of *Ferulago* and confirms a similar chemical profile to that previously reported [17].

### 2.2. Minimal Inhibitory and Minimal Lethal Concentrations of Ferulago spp. Essential Oils

The MIC and MLC of the EOs obtained from the species referred to in Table 1 were determined, with *F. setifolia* and *F. silaifolia* being the most active species. Indeed, these species showed very promising antifungal effects, the latter being very effective (Table 2), especially against *M. canis* and *T. rubrum*, with the lowest recorded values of MIC = 50 µg/mL and MLC = 100 µg/mL. Moreover, a fungicidal effect was observed for *T. mentagrophytes* in *F. setifolia* and for *M. gypseum* and *T. mentagrophytes* in *F. silaifolia*, as the MIC and MLC values were equivalent. For the remaining species, MIC values were quite high (≥400 µg/mL), suggesting a very mild antifungal effect. For this reason, these species were disregarded and not considered in the following studies. To further confirm its antifungal potential, we selected *F. silaifolia*, which presented the most promising effects, to perform antibiofilm assays against the most common dermatophyte species, representing the three different genera: *Epidermophyton floccosum*, *Microsporum canis* and *Trichophyton rubrum*, as detailed in the next section.

### 2.3. Effect of Ferulago silaifolia Essential Oil on the Formation of Dermatophyte Biofilms

In the present study, the effect of *F. silaifolia* EO on the formation and disruption of biofilms formed by the three selected fungal strains was assessed.

Considering the effect of the EO on biofilm formation, it is possible to observe a decrease in biofilm biomass and extracellular matrix deposition in the tested strains (Figure 1A–C), especially at the MIC value, as expected. It is notable that *T. rubrum* was the most susceptible strain to the EOs’ activity, being susceptible even at concentrations as low as MIC/4 (Figure 1C). Indeed, the EO was able to significantly decrease extracellular matrix deposition at 12.5 µg/mL (MIC/4), and at 25 µg/mL (MIC/2), it was also effective in reducing biofilm biomass (Figure 1C). On the other hand, *M. canis* was the least susceptible strain, with all parameters exhibiting a slight decrease but without statistical significance between the positive control (DMSO) and the different EO concentrations.

### 2.4. Effect of Ferulago silaifolia Essential Oil on the Disruption of Dermatophyte Pre-Formed Biofilms

The effect of *F. silaifolia* EO on disrupting pre-formed biofilms was also assessed as these structures are highly resistant to both therapy and host defenses. As shown in Figure 2, a significant susceptibility of *T. rubrum* to the EO (Figure 2C) is noteworthy, with a statistically significant reduction seen in extracellular matrix deposition up to MIC/8 (6.25 µg/mL). In addition, *E. floccosum* was quite susceptible at MIC (50 µg/mL) and MIC/2 (25 µg/mL), particularly in the biomass assay (Figure 2A). In contrast, and similarly to the observations in the biofilm formation inhibition assay, *M. canis* was the least susceptible strain to the EO’s activity (Figure 2B).

### 2.5. Morphological Alterations Induced by Ferulago silaifolia Essential Oil on Dermatophytes

Optical microscopy observations were performed following biofilm assays to assess the morphological alterations induced by *F. silaifolia* EO (MIC) on three fungal strains (*Epidermophyton floccosum*, *Microsporum canis* and *Trichophyton rubrum*). As illustrated in Figure 3 and Figure 4, the EO produced significant morphological changes in both biofilm formation and pre-formed biofilms. Regarding biomass (Figure 3), there is a noticeable increase in septate hyphae (arrows) and less dense mycelia, particularly in *T. rubrum* in the presence of the EO; whereas, in the positive control (DMSO), an evident clustering of hyphae and spores in an organized structure was observed. Indeed, the exposure of the fungi to the EO resulted in hypha growth inhibition or destruction with an increase in planktonic fungi. This effect was more evident on biofilm formation compared to pre-formed biofilms, corroborating the results above. Extracellular matrix deposition (Figure 4) was also significantly compromised with very effective results observed once again in *T. rubrum*, in which no extracellular matrix deposition was observed in the presence of the EO.

### 2.6. Effect of Ferulago setifolia and F. silaifolia Essential Oil on Fibrablast Migration

As both *F. setifolia* and *F. silaifolia* EOs were effective in inhibiting dermatophyte growth, their effect on cell migration was also assessed. As shown in Figure 5 and Figure 6, both oils increased cell migration at the highest concentration devoid of toxicity, although statistical significance was only attained for *F. silaifolia* EO at 25 μg/mL (Figure 6B).

## 3. Discussion

Currently, there is significant neglect regarding dermatophytosis, despite the fact that these infections can progress to severe chronic conditions and become invasive, affecting the dermis and, in some cases, internal organs. This is often due to treatment challenges, as conventional antifungal drugs require long-term administration to be effective. Otherwise, the infection tends to relapse, and the fungi develop resistance to antifungal agents. Given this issue, aromatic plants and their extracts, particularly EOs, have emerged as promising candidates for antifungal treatments, due to their high content in bioactive compounds.

Herein, the antifungal effect of EOs obtained from five *Ferulago* species was assessed using both collection and clinical strains of dermatophytes. Regarding the planktonic forms of dermatophytes, both *F. setifolia* and *F. silaifolia* EOs were quite effective, especially against *M. canis*, *T. mentagrophytes* and *T. rubrum,* although the latter was more active in presenting lower MICs. This aligns with previous research indicating that EOs, especially those rich in monoterpene hydrocarbons, exhibit strong antifungal properties. For example, a previous study conducted by Pinto et al. [16], the only one focusing on *Ferulago* and dermatophytes, demonstrated that the EO of *F. capillaris*, which is rich in α-pinene and limonene, exhibited significant antifungal activity against various pathogens, particularly dermatophytes. However, the activity was significantly lower than that observed in the present study.

Another interesting aspect to point out is that, although several studies have reported the antimicrobial effects of *Ferulago* EOs, the majority focus on bacteria [18] and direct comparisons across studies are quite challenging due to the use of different methodologies. On the other hand, regarding fungi, *Candida* species are generally targeted e.g., [14,15], with *C. albicans* being the most studied strain. In this regard, the EO obtained from the roots of *F. campestris*, with high amounts of α-pinene (58.3–75.0%), showed relevant antifungal effects with a MIC value of 78 μg/mL [14].

It is notable that α-pinene is quite relevant in the genus *Ferulago*, as several species are rich in this compound. Other relevant compounds include cis-chrysanthenyl acetate, and 2,3,6-trimethyl benzaldehyde. A literature review compiled by Pandey and colleagues supports that these compounds, among others, have the capacity to disrupt fungal cell membranes, increasing permeability and leading to cell lysis, making these EOs promising antifungal agents [19]. The high antifungal activity of *F. silaifolia* is likely attributed to the presence of major constituents such as α-pinene, which has been shown to have a potent effect on dermatophytes [20] as well as other fungal strains [21,22]. Additionally, synergistic interactions among various compounds, including minor ones, may also play a role in enhancing the overall activity of the EO and, therefore, cannot be disregarded.

Regarding biofilms, for most strains, an inhibitory effect of *F. silaifolia* EO on biofilm formation was observed, with greater reductions in biofilm biomass and extracellular matrix at MIC concentrations and, in some cases, even at lower concentrations. However, this trend was not so evident in pre-formed biofilms, as these tend to be more resistant. Nevertheless, for *T. rubrum*, a significant reduction in extracellular matrix deposition was observed at very low concentrations, with a statistical significance at MIC/8 (6.25 µg/mL). These results were supported by microscopic observations, which revealed notable morphological changes in biofilms treated with this EO, namely, an evident reduction in hyphae and mycelia density. Also, it was evident that the EO disrupted the biofilm structure, causing the fungal cells to disperse into the surrounding medium. These structural disruptions are critical as they compromise the integrity and functionality of the biofilm, making it more susceptible. In general, significant inhibitions of fungal growth and biofilm, particularly in *T. rubrum*, in the presence of *F. silaifolia* EO are noteworthy and given the importance of biofilms in enhancing fungal survival and resistance to therapy and host defenses, this finding is particularly significant. By eradicating this critical virulence factor, it is possible to control the pathogenesis of dermatophytes. Importantly, this study is the first to report the effect of *Ferulago* EOs on dermatophytes biofilms, providing a foundational basis for future studies. Indeed, studies on the effect of EOs on dermatophyte virulence factors are quite scarce. However, a recent study by Alves-Silva et al. provided new insights into the effects of EOs on dermatophytes, demonstrating that the EO of *Lavandula multifida* has a strong inhibitory effect on the formation of biofilms of dermatophytes and *Candida albicans* [23]. Additionally, the EO was effective in eradicating pre-formed biofilms in all tested strains, with dermatophytes showing greater susceptibility compared to *C. albicans*. Notably, the species *E. floccosum*, *T. mentagrophytes*, *M. gypseum*, and *T. rubrum* exhibited the highest sensitivity. These results support the strong antibiofilm potential of some EOs, endorsing their use in the development of medicinal plant-based antifungal products.

In the present study, *F. silaifolia* was also effective in promoting wound closure at 25 μg/mL. As far as we know, this is the first study to assess the effect of *Ferulago* EOs on fibroblast migration. However, a previous study highlighted the enzyme inhibitory potential, relevant in wound healing, of ethanolic extracts of *Ferulago* spp. with *F. macrosciadia* and *F. syriaca* exhibiting the most potent inhibitory effects on collagenase and elastase enzymes [24].

## 4. Materials and Methods

### 4.1. Plant Material and Essential Oils

Plants were collected from the locations summarized in Table 3 and identified by Prof. Dr. Hayri Duman from Gazi University’s Faculty of Science, Department of Biology. Voucher specimens were deposited at the Herbarium of Ankara University, Faculty of Pharmacy (AEF) with the accession numbers referred to in Table 3.

The EOs were isolated from the fruits of different *Ferulago* species and analyzed using gas chromatography and gas chromatography associated with mass spectrometry, as previously described [17]. Dilutions of each EO were prepared in dimethyl sulfoxide (DMSO) and added to the respective experimental conditions to achieve final concentrations ranging from 50 to 800 μg/mL, with the DMSO concentration in the experimental setups not exceeding 1%.

### 4.2. Antifungal Effect of Essential Oils on Planktonic Fungi

The antifungal activity of *Ferulago* EOs was assessed against dermatophytes from the Colección Española de Cultivos Tipo (CECT), namely, *Microsporum gypseum* CECT 2908, *Trichophyton mentagrophytes* var. *interdigitale* CECT 2958, *T. rubrum* CECT 2794 and *T. verrucosum* CECT 2992, as well as three clinical strains, which are part of the fungal collection of the laboratory: *Epidermophyton floccosum* FF9, *M. canis* FF1 and *T. mentagrophytes* FF7, isolated from patients’ nails and skin.

First, the fungal strains were cultured on Sabouraud dextrose agar (SDA) to ensure optimal growth characteristics and purity. To determine the minimal inhibitory concentration (MIC) and the minimal lethal concentration (MLC) of the oils, the macrodilution method was used, according to Clinical and Laboratory Standards Institute guidelines M38-A2 for filamentous fungi [25]. Dilutions of each EO were prepared in DMSO (50 to 800 μg/mL). Then, using recent cultures of each strain, a fungal suspension was prepared and adjusted to 1 McFarland. The test tubes, containing the respective inoculum and EO concentration, were incubated aerobically at 30 °C for 7 days and MICs were determined by visualizing the turbidity of each test tube. The MIC was determined as the lowest concentration of EO capable of inhibiting the growth of the fungus. After MIC readings, aliquots (20 μL) of broth were taken from each negative tube to evaluate MLCs. Plates containing SDA medium and the inoculated aliquots were incubated at 30 °C for 7 days. The MLC was considered to be the lowest concentration of EO that, after incubation, showed no fungal growth on the plate. For each strain tested, the growing conditions and the sterility of the medium were checked (negative control). The innocuity of DMSO (positive control) was also checked at the highest concentration tested. All experiments were performed in triplicate and repeated if the results differed.

### 4.3. Evaluation of Essential Oils’ Effect on Biofilm Formation

After MICs and MLCs of the fungi were determined, the most promising *Ferulago* species was chosen to evaluate the effect of its EO on the formation and disruption of biofilms formed by the most common species to cause infection, namely, *M. canis*, *E. floccosum* and *T. rubrum*. To evaluate the effect of the EO on the formation of dermatophyte biofilms, 100 μL of conidia suspension adjusted to 1 McFarland was added to sterile 96-well flat-bottomed polystyrene microtiter plates and incubated at 37 °C for 3 h for conidia to adhere. The saline was then removed, and the wells were washed with PBS (0.8% NaCl, 0.02% KH_2_PO_4_. 0.31% Na_2_HPO_4_∙12 H_2_O and 0.02% KCl; pH 7.4) to remove non-adherent cells. Finally, 200 μL of RPMI medium containing different concentrations of the selected EO were added and incubated at 37 °C for 72 h. A negative control containing non-inoculated medium and a positive control with 1% DMSO in inoculated medium (without EO) were also included. To evaluate the effect of the EOs on the formation of dermatophyte biofilms, both biofilm mass and extracellular matrix deposition were quantified, in the presence of EOs’ MIC and lower concentrations (up to MIC/8), using crystal violet and safranin assays, respectively.

Biofilm mass:

After discarding the medium, the biofilms were washed with PBS to remove the non-adherent cells. Then, to fix the biofilms, methanol was added for 10 min, followed by 100 μL of a 0.5% crystal violet solution for 15 min. After removing the solution, the biofilms were washed twice with sterile water and 150 μL of 33% acetic acid was added to dissolve the crystals. Finally, the volume was transferred to another well and the absorbance (*Abs*) was measured at 620 nm [26].

Extracellular matrix:

After discarding the medium, the biofilms were washed with PBS to remove the non-adherent cells. Then 100 μL of a 0.5% safranin red solution was added for 5 min. After removing the solution, the biofilms were washed twice with sterile PBS to remove the unbound dye. Then, 150 μL of 33% acetic acid was added to dissolve the crystals and, finally, the volume was transferred to a new well and the absorbance was measured at 520 nm [27].

### 4.4. Evaluation of Essential Oils’ Effect on Biofilm Disruption

To evaluate the effect of the EO on pre-formed dermatophyte biofilm disruption, suspensions of dermatophyte conidia (200 μL) adjusted to 1 McFarland were added to sterile 96-well flat-bottomed polystyrene microtiter plates and incubated at 37 °C for 3 h to allow the conidia to adhere. After removing the saline, the cells were washed with PBS to remove the non-adherent cells and 200 μL of sterile RPMI medium was added to the plates and incubated at 37 °C for 72 h. The medium was removed, followed by a washing step with PBS. Finally, 200 μL of RPMI with different concentrations of EO (6.25–50 μg/mL) was added to each well and incubated at 37 °C for 24 h. A negative control containing non-inoculated medium and a positive control with 1% DMSO in inoculated medium (without EO) were also included. To evaluate the effect of EO on disrupting the dermatophyte’s biofilm, both biofilm mass and extracellular matrix deposition were quantified, as referred to in Section 4.3.

### 4.5. Effect of the Essential Oil on Fungal Morphology

Fungal morphological alterations in the presence of the most effective EO were evaluated using brightfield microscopy (Motic AE 2000) with a 20× objective (LWD 20×/0.3 Ph).

### 4.6. Cell Viability Assay

The two most effective EOs in inhibiting dermatophytes growth were used to assess their effect on the viability of NIH/3T3 fibroblasts (mouse embryonic fibroblast cell line—ATCC CRL-1658), prior to investigating their impact on cell migration. For that, 50,000 cells/mL were plated on 48-well microplates in DMEM (Dulbecco’s Modified Eagle Medium) with 10% FBS and 1% Penicillin/Streptomycin and allowed to stabilize for 12 h in a culture chamber (37 °C, 5% CO_2_). Then, different concentrations of the EOs (1.562–800 µg/mL) were added to the respective wells for 24 h, after which the medium was removed and fresh medium containing resazurin (1:10) was added for 4 h. The absorbance at 570 nm with a reference filter of 620 nm was registered in an automated plate reader (SLT, Wels, Austria). Cell viability was determined using the following equation:*Cell viability* (%) = *Abs T/Abs Ct* × 100
where *Abs T* is the absorbance (difference between 570 and 620 nm) in the different experimental conditions and *Abs Ct* is the absorbance in the control cells (no EO added).

### 4.7. Scratch Wound Assay

Cell migration was assessed using a scratch wound assay, as previously reported. Briefly, fibroblasts were plated at 300,000 cells/mL in 12-well plates and allowed to grow for 12 h. Then, a scratch was induced with a 200 μL pipette tip, and the detached cells were washed off with PBS. New culture medium with or without the EOs (6.25–25 μg/mL) was added and images were acquired at 0h and 18h post-scratch using a phase-contrast Zeiss Axio HXP IRE 2 microscope (Carl Zeiss AG, Jena, Germany) at 10× magnification (EC Plan-Neofluar 10×/0.3 Ph1) [28,29]. The wound area was measured using ImageJ/Fiji v1.54p software and the results were obtained using the following equation:*Wound closure* (%) = *A* 0h − *A* xh/*A* 0h *×* 100
where *A* 0h is the area of the wound 0h after the scratch and *A* xh is the area 18h post-scratch.

### 4.8. Statistical Analysis

All statistical analysis were performed using GraphPad Prism version 9.5.0 (GraphPad Software, San Diego, CA, USA) and the results are presented as mean values ± SEM (standard error of the mean) from at least three independent experiments performed in duplicate. Statistical significance was determined using a two-way ANOVA followed by Šídák’s or Dunnet’s multiple comparison test. Results with a *p* ≤ 0.05 were considered statistically significant.

## 5. Conclusions

This study investigated the antifungal potential of EOs extracted from *Ferulago* species from Türkiye in inhibiting and disrupting biofilms formed by dermatophytes. The EO of *F. silaifolia* demonstrated significant antifungal activity, particularly against *M. canis* and *T. rubrum*, with MIC values of 50 µg/mL. Notably, *T. rubrum* was the most susceptible fungal species, exhibiting marked morphological changes, including reduced hyphal growth and mycelial density. These findings are promising and suggest that *Ferulago* EOs could serve as alternative antifungal therapies by complementing or replacing conventional treatments while reducing the dosage of synthetic drugs and their associated side effects. This study is the first to highlight the potential of *Ferulago* EOs as effective antifungal agents against dermatophytes, particularly in targeting biofilms, offering a novel approach to treating resistant fungal infections. Moreover, their wound-healing potential adds additional relevance as it may help avoid systemic infection.

## Figures and Tables

**Figure 1 antibiotics-14-00343-f001:**
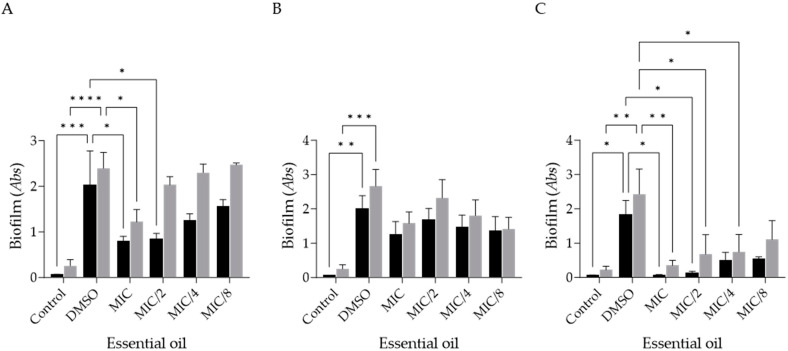
*Ferulago silaifolia* essential oil’s effect on biofilm formation mass (represented by dark bars) and extracellular matrix (represented by grey bars) of *Epidermophyton fluccosum* (**A**), *Microsporum canis* (**B**) and *Trichophyton rubrum* (**C**). Results expressed as absorbance (*Abs*) values and comparisons relative to DMSO (mean ± SEM values; * *p* < 0.05, ** *p* < 0.01, *** *p* < 0.001, **** *p* < 0.0001). DMSO—positive control containing EO-free inoculated medium; control—negative control containing non-inoculated medium.

**Figure 2 antibiotics-14-00343-f002:**
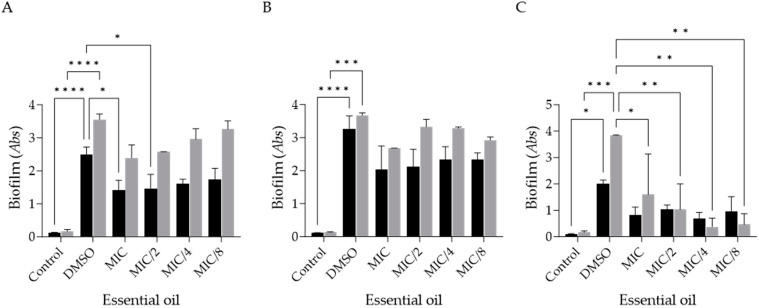
*Ferulago silaifolia* essential oil’s effect on pre-formed biofilm mass (represented by dark bars) and extracellular matrix (represented by grey bars) of *Epidermophyton fluccosum* (**A**), *Microsporum canis* (**B**) and *Trichophyton rubrum* (**C**). Results expressed as absorbance (*Abs*) values and comparisons relative to DMSO (mean ± SEM values; * *p* < 0.05, ** *p* < 0.01, *** *p* < 0.001, **** *p* < 0.0001). DMSO—positive control containing EO-free inoculated medium; control—negative control containing non-inoculated medium.

**Figure 3 antibiotics-14-00343-f003:**
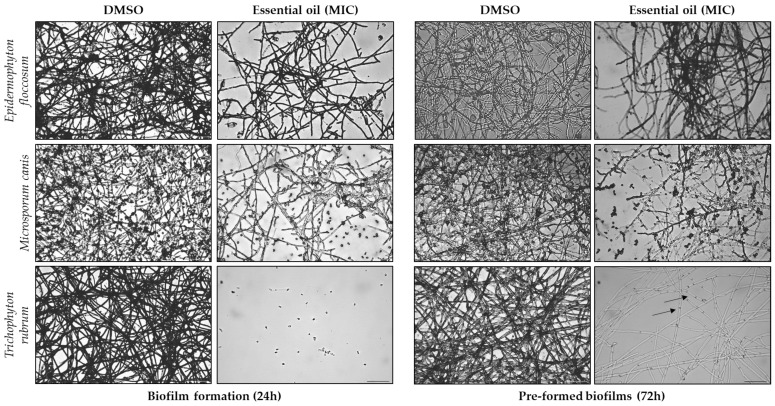
Optical microscopy observations of biofilm formation and pre-formed biofilms in *Epidermophyton floccosum*, *Microsporum canis* and *Trichophyton rubrum*, both untreated (DMSO) and treated with the MICs of *F. silaifolia* essential oil, following biomass quantification using a crystal violet assay. Arrows indicate septate hyphae. Scale bar = 100 μm.

**Figure 4 antibiotics-14-00343-f004:**
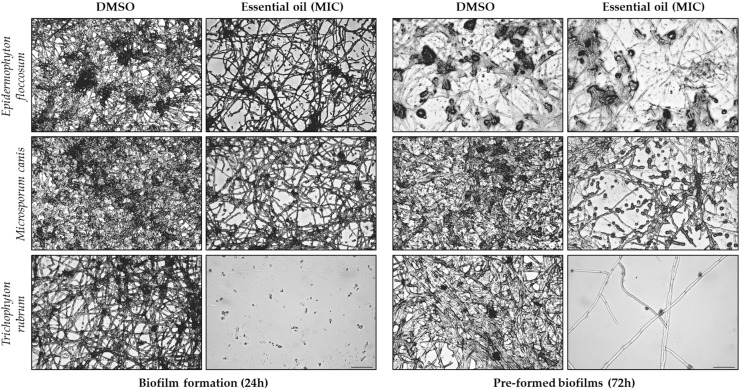
Optical microscopy observations of biofilm formation and pre-formed biofilms in *Epidermophyton floccosum*, *Microsporum canis* and *Trichophyton rubrum*, both untreated (DMSO) and treated with the MICs of *F. silaifolia* essential oil, following biomass quantification using a safranin assay. Scale bar = 100 μm.

**Figure 5 antibiotics-14-00343-f005:**
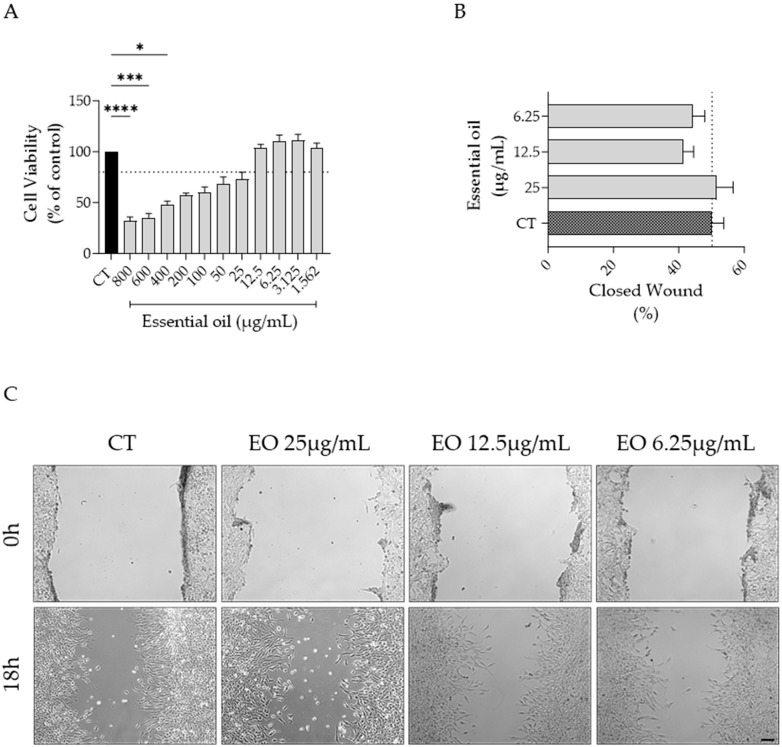
NIH/3T3 fibroblasts migration assessed using a scratch wound healing assay. (**A**) Cell viability following 24h of treatment with *F. setifolia* essential oil. (**B**) Percentage of closed wound at 18h post-scratch; the dashed line represents the percentage of wound closure in control (CT) cells. (**C**) Representative bright-field images of NIH/3T3 fibroblasts 0h and 18h post-scratch. Results as mean ± SEM of three independent assays performed in duplicate. * *p* < 0.05, *** *p* < 0.001 and **** *p* < 0.0001, compared to control (CT). Scale bar = 100 μm.

**Figure 6 antibiotics-14-00343-f006:**
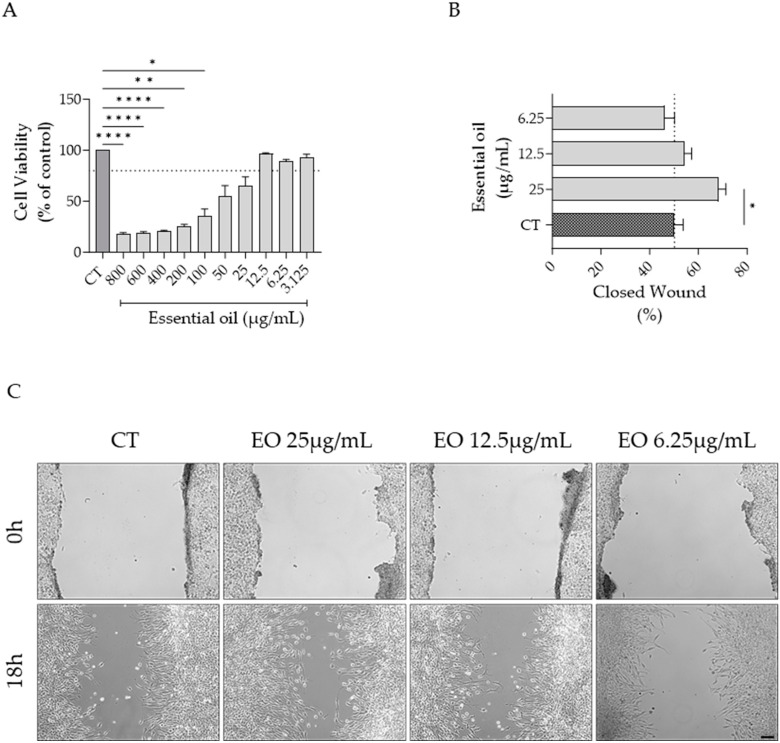
NIH/3T3 fibroblasts migration assessed by the scratch wound healing assay. (**A**) Cell viability following 24 h of treatment with *F. silaifolia* essential oil. (**B**) Percentage of closed wound at 18 h post-scratch; the dashed line represents the percentage of wound closure in control (CT) cells. (**C**) Representative bright-field images of NIH/3T3 fibroblasts 0h and 18h post-scratch. Results as mean ± SEM of three independent assays performed in duplicate. * *p* < 0.05, ** *p* < 0.01 and **** *p* < 0.0001, compared to control (CT). Scale bar = 100 μm.

**Table 1 antibiotics-14-00343-t001:** Main compounds of *Ferulago* spp. essential oil.

Species	Composition	
	Main Compounds	Amount (%)
*F. cassia* Boiss.	α-pinene	61.0
	cis-crysanthenyl acetate	21.3
	α-phellandrene	4.5
*F. isaurica* Peşmen	α-pinene	17.3
	α-phellandrene	36.5
	limonene	16.3
*F. setifolia* K. Koch	α-pinene	39.1
	2,3,6-trimethyl benzaldehyde	35.1
	sabinene	8.2
*F. silaifolia* (Boiss.) Boiss.	α-pinene	45.4
	cis-crysanthenyl acetate	39.1
	myrcene	1.7
*F. syriaca* Boiss.	myrcene	16.8
	α-phellandrene	8.8
	cubenol	9.1

**Table 2 antibiotics-14-00343-t002:** Antifungal effect of *Ferulago silaifolia* and *F. setifolia* essential oils.

Dermatophyte Strains	*Ferulago silaifolia*		*Ferulago setifolia*	
	MIC	MLC	MIC	MLC
*Epidermophyton floccosum* FF9	100	100–200	100–200	200–400
*Microsporum canis* FF1	50	100	200	200–400
*Microsporum gypseum* CECT 2908	200	200	200	400–600
*Trichophyton mentagrophytes* FF7	100–200	100–200	200	200
*Trichophyton mentagrophytes* var. *interdigitale* CECT 2958	200	400	200	600
*Trichophyton rubrum* CECT 2794	50	100	50–100	100
*Trichophyton verrucosum* CECT 2992	200–400	600	200	400

MIC—Minimal Inhibitory Concentration; MLC—Minimal Lethal Concentration; MIC and MLC determined by the macrodilution method and expressed in μg/mL.

**Table 3 antibiotics-14-00343-t003:** Date and site of collection of *Ferulago* species.

Species	Local Name	Collection Date	Site of Collection	Accession Number
*F. cassia* (1)	şeytankişnişi	20 June 2016	Konya: Beyşehir, Tınaztepe-Bozkır old road, 1st km, serpentine areas, in gladelike areas under *Pinus nigra*, 1555 m	AEF 28775
*F. isaurica* (2)	kargıkişnişi	21 June 2016	Antalya-Alanya, between Durbannaz-Banlıca, under *Pinus brutia* forest, calcaerous rocks, 837 m	AEF 28778
*F. setifolia* (3)	kılkişniş	14 July 2017	Erzincan: Üzümlü district, Karakaya Town, between Karakaya-Tekçam Highland, high mountain steppe, 2047 m	AEF 28770
*F. silaifolia* (4)	ulukişniş	7 June 2016	Bursa, Uludağ road, 100 km to National Park, under *Castanea sativa* Mill. and *Pinus nigra* forest, 837 m	AEF 28771
*F. syriaca* (5)	kırkişnişi	22 June 2016	Hatay: Harbiye-Şenköy roa, ca. 5th–6th km, around maquis, 452 m	AEF28779

## Data Availability

The data presented in this study are available on request from the corresponding author.

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
