# Peer review of "Unlocking the Skin-Protective Effects of *Ferulago* spp. Essential Oils: A Focus on Antidermatophytic and Wound-Healing Potential"

_antibiotics, 2025, doi:10.3390/antibiotics14040343_

Round 1
Reviewer 1 Report
Comments and Suggestions for Authors
The study entitled “Unlocking skin protective effects of Ferulago spp. essential oils: a focus on antidermatophytic and wound healing potentials” by Kılıç, C and collaborators presents an important and promising approach regarding the antifungal potential of essential oils. Congratulations to the authors for the beautiful work. I have few considerations. Follow...
Abstract
Lines 43-44: The authors should only suggest that the species has healing potential due to the promising result of the cell migration test, not affirm, as the cell migration test alone is not sufficient for the affirmation.
Introduction
Line 48- Set the spacing for the first line of the paragraph.
Lines 48-73: Long paragraph, reduce.
Lines 94-96: Paragraph too short in relation to the others, correct.
Results
Set the spacing of the first line of all paragraphs of the texts in the results section.
Lines 134-137: The text reflects discussion. In this section, only the results are presented.
Lines 139-141: This text is methodology, there is no need to be in results.
Line 153, Figure 1: Place images A, B, and C side by side similar to Figures 5 and 6.
Line 168, Figure 2: Place images A, B, and C side by side similar to Figures 5 and 6.
Lines 198-201: The text reflects discussion. In this section, present only the results.
Line 204: Add the period.
Discussion
Insert the spacing of the first line of the paragraph throughout the discussion text.
Materials and Methods
Line 288: Insert the spacing of the first line of the paragraph.
Lines 292-294: The methodology should be written so that the reader can read your work and be able to reproduce the methodology without having to look for the cited reference.
Line 304: Shouldn't microdilution be correct instead of macrodilution?
Lines 362-363: Add the reference.
Lines 365-377: Add the reference.
Lines 379-388: Add the reference
Line 394: p value < 0.05 should be p ≤ 0.05
Line 407-408: Only suggest the potential for healing, do not affirm it.
References
Format all references according to the journal's standards
Author Response
Comments and Suggestions from Reviewer #1
The study entitled “Unlocking skin protective effects of Ferulago spp. essential oils: a focus on antidermatophytic and wound healing potentials” by Kılıç, C and collaborators presents an important and promising approach regarding the antifungal potential of essential oils. Congratulations to the authors for the beautiful work. I have few considerations.
- Abstract
Lines 43-44: The authors should only suggest that the species has healing potential due to the promising result of the cell migration test, not affirm, as the cell migration test alone is not sufficient for the affirmation.
Thank you for the suggestion. We agree and have modified the final sentence of the abstract.
- Introduction
Line 48- Set the spacing for the first line of the paragraph.
Thank you for the suggestion. The spacing was included.
Lines 48-73: Long paragraph, reduce.
We acknowledge the suggestion and have divided the paragraph in parts.
Lines 94-96: Paragraph too short in relation to the others, correct.
We agree. This paragraph was joined with the following
- Results
Set the spacing of the first line of all paragraphs of the texts in the results section.
Thank you for this notice. We have included a spacing in the first line of all paragraphs in the results section.
Lines 134-137: The text reflects discussion. In this section, only the results are presented.
We appreciate the comment and have removed the text that reflects discussion.
Lines 139-141: This text is methodology, there is no need to be in results.
We acknowledge the comment and have removed the text that reflects methodology.
Line 153, Figure 1: Place images A, B, and C side by side similar to Figures 5 and 6.
Thank you for the suggestion. We have reorganized the figure accordingly.
Line 168, Figure 2: Place images A, B, and C side by side similar to Figures 5 and 6.
Thank you for the suggestion. We have reorganized the figure accordingly.
Lines 198-201: The text reflects discussion. In this section, present only the results.
We appreciate the comment and have removed the text that reflects discussion.
Line 204: Add the period.
We apologize for this slight mistake. A period was included at the end of the sentence.
- Discussion
Insert the spacing of the first line of the paragraph throughout the discussion text.
Thank you for this notice. We have included a spacing in the first line of all paragraphs in the discussion section.
- Materials and Methods
Line 288: Insert the spacing of the first line of the paragraph.
Thank you. We have included a spacing in the first line of all paragraphs in the materials and methods section.
Lines 292-294: The methodology should be written so that the reader can read your work and be able to reproduce the methodology without having to look for the cited reference.
Thank you for the comment. We have completed the methodology section with more information.
Line 304: Shouldn't microdilution be correct instead of macrodilution?
This is not a mistake. In fact, the MIC and MLC values in this study were determined using a macrodilution method and not a microdilution one.
Lines 362-363: Add the reference.
Thank you for the suggestion; however, this methodology was not based on any reference and therefore it was not incuded.
Lines 365-377: Add the reference.
We agree and have included a reference for this methodology.
Lines 379-388: Add the reference
We agree and have included a reference for this methodology.
Line 394: p value < 0.05 should be p ≤ 0.05
Thank you. We have performed the suggested correction.
Line 407-408: Only suggest the potential for healing, do not affirm it.
Thank you for the suggestion. We have added the word potential to suggest the effect and nor affirm it.
- References
Format all references according to the journal's standards
We have checked and the reference style matches the one recommended by the journal.
Reviewer 2 Report
Comments and Suggestions for Authors
The study of Kiliç and colleagues reports the antidermatophytic and wound healing potentials of essential oil (EO) from Ferulago species. In general, the manuscript is technically sound. However, some issues need to be clarified/amended.
Methods
-Authors should make it clear in their methodologies who is the positive and negative control. For example, in the description of the antibiofilm activity there is no mention of DMSO, and in the representation of the results a column referring to DMSO is presented. I recommend adding a sentence to item 4.1 explaining how the essential oils were solubilized for the tests.
-Lines 306-307: “Serial doubling dilutions of each EO were prepared in dimethyl sulphoxide (DMSO), with different concentrations ranging from 50 to 800 μg/mL.” The EOs were first diluted in pure DMSO and then added to the culture medium? Please clarify.
-Please add a reference for the use of safranin for detection of extracellular matrix in biofilms.
Results
-Table 2, please verify:
Replace the numbers 3 and 4 in the table with the name of the plant species.
Why are two MIC values shown for some fungal species? The authors must present the MIC values for all the EOs tested.
-Why was only one species of Ferulago evaluated for antibiofilm activity? EOs that have no inhibitory effect on planktonic cells, or exhibit good MIC values, may have a good antibiofilm effect. And as the authors state that the antibiofilm activity of the EOs of these plants is little explored, it would be a good opportunity to make the results available to the public.
-Figures 1 and 2: authors should replace the names of the fungal species on the y-axis with what is represented. I recommend showing images at higher magnifications to make it easier to see the changes described in the text.
-The term “immature biofilm” is not appropriate, as well as “mature biofilm”, since no study of biofilm formation kinetics was carried out in the study. I recommend using “effect of EO on biofilm formation” and “72 h-biofilms” (or pre-formed biofilm).
-Lines 181-182: “Indeed, the exposure of the fungi to the EO resulted in hypha growth inhibition or destruction with an increase in planktonic fungi”. Please explain why is there an “increase in planktonic fungi”?
-Figures 3 and 4: please improve the presentation of the scale bar. It is difficult to see.
-Apparently F. setifolia does not seem to exert a proliferative effect in the scratch assay. The percentage of cell migration was similar to the untreated control. Please verify.
-Figures 5 and 6: please add the meaning of the dashed lines. A scale bar or magnification for the images in the C figures should be added.
Discussion
-The authors should further discuss the inhibitory effect, especially on planktonic cells, related to the chemical composition of EOs. For example, according to table 1, F. cassia and F. silaifolia have the same major constituents. Why were there differences in MIC values?
-Minor comments:
Please revise the fungal scientific names, including in Figures. (Epidermothyton), and make minor corrections to the English language.
Comments on the Quality of English LanguageMinor corrections to the English language.
Author Response
Comments and Suggestions from Reviewer #2
The study of Kiliç and colleagues reports the antidermatophytic and wound healing potentials of essential oil (EO) from Ferulago species. In general, the manuscript is technically sound. However, some issues need to be clarified/amended.
- Methods
Authors should make it clear in their methodologies who is the positive and negative control. For example, in the description of the antibiofilm activity there is no mention of DMSO, and in the representation of the results a column referring to DMSO is presented. I recommend adding a sentence to item 4.1 explaining how the essential oils were solubilized for the tests.
We acknowledge the suggestion. To avoid confusion, we have include a sentence on this matter in section 4.1.
Lines 306-307: “Serial doubling dilutions of each EO were prepared in dimethyl sulphoxide (DMSO), with different concentrations ranging from 50 to 800 μg/mL.” The EOs were first diluted in pure DMSO and then added to the culture medium? Please clarify.
Thank you once again. We have modified and completed this information in all sections that mention essential oils dilutions.
Please add a reference for the use of safranin for detection of extracellular matrix in biofilms.
We acknowledge the suggestion. A reference for the detection of both biofilm mass and extracellular matrix was included.
- Results
Table 2, please verify:
Replace the numbers 3 and 4 in the table with the name of the plant species.
Thank you for the suggestion. We have replaced the numbers by the species names.
Why are two MIC values shown for some fungal species? The authors must present the MIC values for all the EOs tested.
Our antifungal assay is performed in duplicate and repeated at least three times. If the results are not consistent, two additional repetitions are conducted to confirm the data. When the results fall between two consecutive concentrations, we report a range of values instead of a single concentration point. This ensures greater accuracy in interpreting the results.
Why was only one species of Ferulago evaluated for antibiofilm activity? EOs that have no inhibitory effect on planktonic cells, or exhibit good MIC values, may have a good antibiofilm effect. And as the authors state that the antibiofilm activity of the EOs of these plants is little explored, it would be a good opportunity to make the results available to the public.
Ferulago silaifolia was chosen not only because it demonstrated more effective Minimum Inhibitory Concentrations (MICs) and Minimum Lethal Concentrations (MLCs) but also because it is an endemic species in Türkiye and grows in a limited part of the country.
Figures 1 and 2: authors should replace the names of the fungal species on the y-axis with what is represented. I recommend showing images at higher magnifications to make it easier to see the changes described in the text.
The Y-axis label was modified to provide more clarity. Since both biomass and extracellular matrix are represented, we chose to use the percentage of biofilm to encompass both experiments. Regarding the higher magnification images, Reviewer #1 requested that they be placed side by side, similar to Figures 5 and 6. This modification has been made, and the font size has been increased for better visibility.
The term “immature biofilm” is not appropriate, as well as “mature biofilm”, since no study of biofilm formation kinetics was carried out in the study. I recommend using “effect of EO on biofilm formation” and “72 h-biofilms” (or pre-formed biofilm).
We acknowledge your concern and have modified these terms in the text and figures.
Lines 181-182: “Indeed, the exposure of the fungi to the EO resulted in hypha growth inhibition or destruction with an increase in planktonic fungi”. Please explain why is there an “increase in planktonic fungi”?
In our experiments, it was evident that the essential oil disrupted the biofilm structure, leading to the detachment of fungal cells, which eventually became planktonic (free-floating). This suggests that the essential oil weakens the biofilm, causing the fungal cells to disperse into the surrounding medium. This information was included in the discussion section.
Figures 3 and 4: please improve the presentation of the scale bar. It is difficult to see.
The images were improved. The scale bar is now more evident and for uniformization purposes E. fluccosum was placed first.
Apparently F. setifolia does not seem to exert a proliferative effect in the scratch assay. The percentage of cell migration was similar to the untreated control. Please verify.
Figures 5 and 6: please add the meaning of the dashed lines. A scale bar or magnification for the images in the C figures should be added.
In fact, F. setifolia essential oil does not improve wound healing. The meaning of the dashed line was included in figures legend as well as the magnification of figure c.
- Discussion
The authors should further discuss the inhibitory effect, especially on planktonic cells, related to the chemical composition of EOs. For example, according to table 1, F. cassia and F. silaifolia have the same major constituents. Why were there differences in MIC values?
Thank you for your comments. We have revised the discussion section to include the observed increase in planktonic cells. Regarding the composition of the essential oils, we have updated Table 1 to include the next major compound in these two species. Moreover, although the oils contain similar major compounds, they differ in concentration and the role of minor compounds, which may have synergistic effects contributing to the overall activity, should not be overlooked. This information is also in the discussion section.
Minor comments:
Please revise the fungal scientific names, including in Figures. (Epidermothyton), and make minor corrections to the English language.
Thank you. We have performed the corrections.
Comments on the Quality of English Language
Minor corrections to the English language.
Thank you. We have read the manuscript carefully and corrected some mistakes.
Round 2
Reviewer 2 Report
Comments and Suggestions for Authors
Dear Authors, thank you for all replies.
There are minor corrections to the revised manuscript, which can be made at the proofing stage.
Table 2: please remove “and” before “essential oils”.
Line 240: please correct “releavant”
Line 279: please add italic to “F. macrosciadia”
Line 300: In order not to cause ethical problems, add that the clinical isolates are part of the fungal collection of the laboratory, or of the place that donated the isolates.
Line 302: please correct “determinate”
Kind regards.
Author Response
There are minor corrections to the revised manuscript, which can be made at the proofing stage.
Thank you so much for the careful review of our paper.
Table 2: please remove “and” before “essential oils”
The correction was performed.
Line 240: please correct “releavant”
Thank you. The mistake was corrected.
Line 279: please add italic to “F. macrosciadia”
The species was written in italic.
Line 300: In order not to cause ethical problems, add that the clinical isolates are part of the fungal collection of the laboratory, or of the place that donated the isolates.
We acknowledge your concern and have included the suggested information.
Line 302: please correct “determinate”
The mistake was corrected.